# Exploring Simple, High Quality Out-of-Distribution Detection with L2 Normalization

**Jarrod Haas**                                                          *jhaas@sfu.ca*
*SARlab, Department of Engineering Science*
*Simon Fraser University*

**William Yolland**                                                      *yollandw@gmail.com*
*MetaOptima*

**Bernhard Rabus**                                                       *bernhard_t_rabus@sfu.ca*
*SARlab, Department of Engineering Science*
*Simon Fraser University*

**Reviewed on OpenReview:** *https://openreview.net/forum?id=daX2UkLMS0*

## Abstract

We demonstrate that L2 normalization over feature space can produce capable performance for Out-of-Distribution (OoD) detection for some models and datasets. Although it does not demonstrate outright state-of-the-art performance, this method is notable for its extreme simplicity: it requires only two addition lines of code, and does not need specialized loss functions, image augmentations, outlier exposure or extra parameter tuning. We also observe that training may be more efficient for some datasets and architectures. Notably, only 60 epochs with ResNet18 on CIFAR10 (or 100 epochs with ResNet50) can produce performance within two percentage points (AUROC) of several state-of-the-art methods for some near and far OoD datasets. We provide theoretical and empirical support for this method, and demonstrate viability across five architectures and three In-Distribution (ID) datasets.

## 1 Introduction

Neural networks give unreliable confidence scores when processing images outside their training distributions, hindering applications in real-world settings where reliable failure modes are critical (Rabanser et al., 2018; Chen et al., 2020; Henne et al., 2020). Much work has been done to address this shortcoming. Although Monte-Carlo Dropout (MCD) and model ensembles were initially popular (Gal and Ghahramani, 2016; Lakshminarayanan et al., 2017), they have quickly become obsolete. A large, diverse landscape of approaches has replaced these simpler and well-studied methods, and has improved OoD detection substantially (see Section 2.2). While recent improvements in detection performance have been remarkable, the complexity of many methods can be intimidating, and it is not always clear whether leaderboard improvements are due to external factors such as introducing new or enlarging existing models, hyperparameters, training schedules or augmentation schemes. This difficulty in assessing real progress is not unique to the OoD research community, and has already emerged as a concern within the metric learning community (Musgrave et al., 2020). Our motivation is therefore to provide a method that 1) is simple, 2) shows promise as a effective and efficient baseline and 3) has theoretical (as well as empirical support) that provides insight that could be useful for further development.

L2 normalization over feature space has been utilized in the facial recognition literature for several years but has only recently been explored more generally with respect to OoD detection (Regmi et al., 2023; Park et al., 2023; Yu et al., 2020) and training efficiency (Haas et al., 2023; Yaras et al., 2023). We build upon this

**Algorithm 1: L2 Normalization of Features**

```
def forward(self, x):
    z = self.encoder(x)
    featurenorm = torch.norm(z).detach().clone()
    z = torch.Functional.normalize(z, p=2, dim=1)
    y = self.fc(z)
    return y, featurenorm
```

Figure 1: A Pytorch code snippet illustrating the proposed method, which only requires the addition of two lines of code to a standard forward function. When training is complete, pre-normalized feature norms can be sampled along with model predictions in the usual manner. These norms are the OoD score: smaller norms are more likely to be OoD.

previous work, demonstrating that this method (which requires only two lines of PyTorch code, see Figure 1), can result in information-rich feature vector norms usable as an OoD scoring method for some architectures and datasets. We observe that feature magnitudes trained without L2 normalization may be not useful in this regard, and provide evidence that this is because cross-entropy (CE) loss under Neural Collapse (NC) promotes equal sized feature norms for all inputs. L2 normalization resolves this by decoupling the magnitude and direction of feature vectors during training. This satisfies optimal NC equinormality conditions while allowing feature norms to be highly variable in size. Borrowing insight from Wang et al. (2017a), we show that norms can grow large due to weight updates during backpropagation. This results in a greater degree of information encoded in feature norms: intuitively, larger norms for a given image during a forward pass at test time indicate that the network is more familiar with that input.

A summary of our contributions is as follows:

- A method for OoD detection on small datasets that requires no additional parameters and works as a simple modification to standard architectures

- A method that is more training efficient for some architectures, most notably ResNet18 and ResNet50, in which cases far less training is required than other recent methods, some of which report several hundred more epochs of training than used for our baseline

- A theoretical rationale for the connection between L2 normalization of features, Neural Collapse and feature magnitudes that we hope can inspire future work in improving how feature information can be used for OoD detection

## 2 Background

### 2.1 Problem Setup

A standard classification model maps images to classes $f : x \to \hat{y}$, where $x = (x_1, ..., x_n) \in X^N$ is the set of $N$ images, and $y = \{1, ..., k\} \in Y^N$ is the set of labels from $k$ classes for these images. The model is composed of a feature extractor (or encoder) $H : x \to \mathbb{R}^d$ which maps images to feature space vectors $z = (z_1, ..., z_n) \in Z^N$, with $d$ being the dimension of the vector. Along with a decision layer $W^{d \times k}$ and a bias term $b$, this becomes an optimization problem of the form:

$$\min_{W,b,z} \frac{1}{N} \sum_N L_{CE}(Wz + b, y_k) \tag{1}$$

where $y_k$ is the label of the correct class, and $L_{CE}$ is the Cross Entropy loss function. Note that the logits $Wz + b$ are transformed by the softmax function for the entropy computation. For analysis, we consider the

| Method | Network | SVHN | CIFAR100 | Accuracy |
|---|---|---|---|---|
| Mahalanobis (Lee et al.) | ResNet34 | 99.1 | 90.5 | N/A |
| LogitNorm+ (Wei et al.) | ResNet18 | 97.1 | 91.0 | 94.4 |
| FeatureNorm (Yu et al.) | ResNet18 | 97.8 | 74.5 | N/A |
| KNN+ (Sun et al.) | ResNet18 | 99.5 | N/A | 95.1 |
| SSD+ (Sehwag et al.) | ResNet50 | 99.9 | 93.4 | 94.4 |
| CSI (Tack et al.) | ResNet18 | 99.8 | 89.2 | 95.1 |
| Gram (Sastry and Oore) | ResNet34 | 99.5 | 79.0 | N/A |
| Contrastive Training (Winkens et al.) | WideResNet50_4 | 99.5 | 92.9 | N/A |
| ERD++ (Tifrea et al.) | 3x ResNet20 | 1.00 | 95.0 | N/A |
| Transformer + Mahalanobis (Fort et al.) | R50+ViT-B_16 | 99.9 | 98.5 | 98.7 |
| L2 Norm 60 | ResNet18 | 97.4 | 88.7 | 92.6 |
| L2 Norm 100 | ResNet50 | 98.6 | 90.4 | 94.1 |

Table 1: AUROC scores for baselines trained on CIFAR10 and tested on far OoD (SVHN) and near OoD (CIFAR100) data sets. Note that ERD++ requires ensembling, the transformer is a far larger model, and contrastive loss methods require additional training, which places these in a different computation class. We include them to show that our method compares favourably when considering efficiency. ID accuracy scores are often not reported for OoD methods. Mahalanobis, FeatureNorm and Gram methods are post-hoc (independent of training) and accuracy would be as expected for the indicated models. ERD++, KNN+, CSI and Contrastive Training all employ contrastive loss which would generally improve accuracy.

encoder $H$ as a black box that produces features $z$ which collapse to a Simplex Equiangular Tight Frame (ETF), as in the Unconstrained Features Model (UFM) (Mixon et al., 2022; Ji et al., 2021). We use VGG16, ResNet18 and ResNet50, ConvNeXt, and Compact Vision Transformers as encoders (See Section A.1 for more details) (Hassani et al., 2021; Liu et al., 2022).

To evaluate models, we merge ID and OoD images into a single test set. OoD performance is then a binary classification task, where we measure how well OoD images can be separated from ID images using a score derived from our model. Our score in this case is the L2 norm of each image's *unnormalized* feature vector $z$, which is input to AUROC and FPR95 scoring functions (see Figure 1).

## 2.2 Related Work

A large number of OoD methods have been published in recent years, and no standard taxonomy for these exists. We situate our work within this rapidly evolving and expanding landscape of research, which includes several categories:

- **Bayesian** - Concerned with approximations of Bayesian statistics to deep learning, such as Monte Carlo Dropout, Probabilistic Back-Propagation, mean-field approximations, variational inference methods

- **Outlier exposure** - Requiring real or synthetic OoD samples during training or tuning of metrics

- **Density/Distance methods** - Analysis of feature space using density estimation (e.g. normalizing flows, Gaussian Mixture Models (Mahalanobis distance), or using non-parametric methods such as euclidean distance or KNNs

- **Generative models** - Use of a learned implicit (e.g. GANs) or explicit (e.g. normalizing flows) distribution over the data set, and exploiting this to evaluate the probability of samples

- **Augmentation** - Changes to input data before model ingestion during training or inference

- **Hyperparameters** - Use of training regimes, loss functions, optimizers etc.

- **Measurement** - Use of parts of the network other than softmax outputs to obtain a confidence score, or making alterations to softmax outputs to obtain a confidence score

- **Architectural** - Changes to standard architectures, such as ResNet or Transformers, to affect how confidence scores are obtained

Several papers falling under multiple of the above categories have explored concepts related to L2 normalization and OoD detection. The face verification community has used L2 normalization over feature space, noting that it can improve accuracy and face-matching quality during training. Ranjan et al. (2017) use a scaled L2 normalization layer over features, but provide no insight as to why this improves performance. Wang et al. (2017b) produce a similar idea, and note that L2 normalization over features can produce arbitrarily large gradients without weight decay (an idea which we borrow), but view this more as a side-effect that needs to be mitigated than a feature. Yu et al. (2020) use L2 norms of features as a score to detect non-face OoD samples, but implements a modified softmax loss with an "uncertainty branch" and offers no insight into why the method works.

Haas et al. (2023) study improvements to the Deep Deterministic Uncertainty benchmark Mukhoti et al. (2021) using L2 normalization of features. They also report an increase in the speed of training and suggest a connection to Neural Collapse, although the connection is not clear. This analysis is limited to a study of density estimation over L2-normed feature space using Gaussian Mixture Models applied to ResNet18 and ResNet50, and does not investigate the utility of feature magnitudes themselves. Their method requires L2 normalization at test time as well as at training time.

Yu et al. (2023) show that using the norm of intermediate network layer outputs (after training) can provide good far-OoD separability (the method does not work well with near-OoD data), and use pixel-wise shuffled training data to assess which layer in a particular network may be most suitable. Wei et al. (2022) achieve good results by L2-normalizing *logits* during training, in conjunction with a temperature parameter. Finally, Regmi et al. (2023) concurrently proposed the same method as ours, and provides results for additional datasets. However, we note that none of these three papers give explanations of why their methods provide improved OoD separability.

Vaze et al. (2022) briefly offer intuition of why feature norms grow larger for ID examples (larger norms lower CE loss values; smaller norms hedge against incorrect predictions in hard or uncertain examples). However, this is is an afterthought in their appendix. Their main contribution is exploiting extreme over-training (600 epochs), augmentation strategies, and learning rate schedules for open-set recognition. We also note that feature norms *do* grow larger for ID examples. However, this effect is subtle and not competitive as a scoring rule on its own, and we explain why theoretically and demonstrate this experimentally.

Park et al. (2023) provide some theoretical analysis as to why feature norms are useful as OoD detectors. Their two main arguments are that feature norms of layers are equivalent to confidence scores, and that feature norms tend to be larger for ID examples, regardless of class. However, their analysis again depends on extreme over training with a cosine annealing schedule (800 epochs for ResNet18 under CE loss), requires the addition of a multi-layer perceptron (MLP) head (with a much larger output size) to the encoder, requires a temperature parameter, and does not connect L2 normalization with neural collapse and feature norm behaviour. We note that adding L2 normalization to the output of an MLP head placed on top of the encoder is a fundamentally different analysis from encoder output that immediately precedes a decision layer (for more details, see Ji et al. (2021), Papyan et al. (2020) and Mixon et al. (2022)).

## 3 Methodology

### 3.1 Cross Entropy Loss and Neural Collapse

Papyan et al. (2020) demonstrated that Neural Collapse occurs for multiple architecture types and datasets. We are concerned with the first two properties of NC:

NC1: Variability collapse: the within-class covariance of each class in feature space approaches zero. In other words, the longer a model is trained, the more it will collapse each input class to a single point in feature space. This measurement is defined as:

$$NC_1 = Tr\{\Sigma_W \Sigma_B^\dagger \backslash C\} \tag{2}$$

where $\Sigma_W$ is the variance within each class, $\Sigma_B$ is the variance between all classes, $Tr$ is the matrix trace operator, $[\cdot]^\dagger$ indicates the Moore-Penrose pseudoinverse, and $C$ is the number of classes (Figure 2).

NC2: Convergence to a Simplex Equiangular Tight Frame (Simplex ETF): the angles between each pair of (collapsed) class means are maximized and equal (equiangularity) and the distances of each class mean from the global mean of classes are equal (equinormality). In other words, class centroids are placed at maximally equiangular locations on a hypersphere.

The equinormality of class means is given by the coefficient of variation of their norms,

$$EN_{means} = \frac{std_c(\parallel u_c - u_G \parallel_2)}{avg_c(\parallel u_c - u_G \parallel_2)} \tag{3}$$

where $u_c, u_G$ in $\mathbb{R}^d$ are the class means and global mean of classes, $std$ and $avg$ are standard deviation and average operators, and $\parallel \cdot \parallel_2$ is the L2-norm operator.

Maximum equiangularity between classes in feature space is measured as:

$$EA_{means} = \frac{\parallel G + C_{cos} - diag\{G + C_{max}\} \parallel_1}{C * (C - 1)} \tag{4}$$

where $G$ is the Gram matrix of normalized class means, $C_{cos}$ is a matrix with elements $-1/(C-1)$, $diag$ returns matrix diagonal, and $\|\cdot\|_1$ is the L1 norm (Figure 2).

Lu and Steinerberger (2020) and Graf et al. (2021) showed that the lower bound on CE loss is directly tied to neural collapse. Equality is possible if and only if features collapse to $k$ equal length, maximally equiangular class vectors and the $k$ decision layer columns in $W$ form a dual space with class vectors up to a scalar:

$$L_{CE}(Wz + b, y_k) \geq log\left(1 + (k-1)\exp(-\rho_z \frac{\sqrt{k}}{k-1}||W||_F)\right) \tag{5}$$

where $||W||_F$ is the Frobenius norm over decision layer weights, $\rho_z \geq 0$, and $||z||_2$ is less than $\rho_z$.

Ji et al. (2021) showed that cross-entropy, without constraints or regularization, is sufficient for convergence to neural collapse under stochastic gradient descent, but this has also been shown under weight decay and $L_P$ norm constraints (E and Wojtowytsch, 2020; Zhu et al., 2021; Yaras et al., 2023). Note that model convergence and total neural collapse are not the same. Models approach complete neural collapse through training, but in some cases training well beyond convergence is required Papyan et al. (2020).

Theoretically, encoders optimized under CE loss will produce feature vectors of equal length and direction for each class. This means that feature norms will tend to contain *class* information at the expense of *image* information (i.e. information about specific features within each image regardless of its class). Intuitively, this is unsurprising given that the classification task optimized under CE loss is simply to categorize images and not to articulate intra-class differences. Still, for some architectures and datasets, it is possible that equinormality constraints may be relaxed while still allowing good generalization. Loss may not reach its absolute minimum due to increases in intra-class variability (NC1) or the relative positions of class means (NC2). This could occur for a number of reasons, including model capacity, dataset complexity, class imbalances, or a high numbers of classes encoded into a relatively low dimensional feature space (Graf

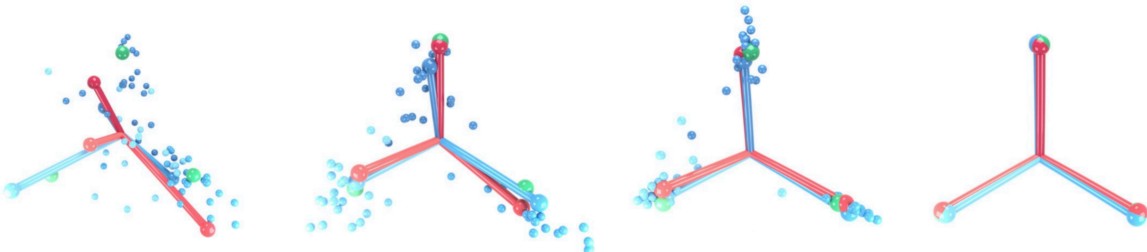

Figure 2: Progression of Neural Collapse during training (left to right). Under Cross Entropy loss, features converge to equinormal and equiangular vectors. Small blue spheres represent extracted features (classes are different shades of blue), blue ball-and-sticks are class-means, red ball-and-sticks are linear classifiers. The simplex ETF pictured is on the 2D plane in 3D space, such that each arm is equidistant at 120 degrees. Image from (Papyan et al., 2020).

et al., 2021). Under these conditions, we speculate that feature norms could be more useful without L2 normalization.

We study cases where feature norms, as a direct measure of input familiarity, can become more useful with L2 normalization. We show that this is at least the case for several architectures trained on CIFAR10, CIFAR100 and TinyImageNet datasets. We note that neural collapse is not *necessary* for our method to work. Rather, our method allows feature norms to be useful as OoD measures *despite* the presence of strong equinormality, when it prevents useful image information from being encoded in feature vectors.

### 3.2 L2 Normalization: Decoupling Feature Vectors from Equinormality

L2 normalization over features is defined as:

$$z_{norm} = \frac{z}{max(\|z\|_2, \epsilon)} \tag{6}$$

where $\epsilon$ is an error term for numerical stability and $z$ is the output of the encoder.

When $L_2$ normalization is applied (during training), the hyper-sphere constraint that exists at optimal CE loss is already met – there is no need for the model to adjust weights in such a manner as to promote equinormality. This allows for much greater variability in feature norms.

Wang et al. (2017b) pointed out that L2 normalization during training had the potentially problematic side-effect of allowing feature magnitudes to grow arbitrarily large in the absence of weight decay. That is, when features are normalized, the gradient of the loss w.r.t. features is orthogonal to the features:

$$\frac{\partial L}{\partial z} = \frac{\partial L}{\partial z_{norm}} \frac{\partial z_{norm}}{\partial z}, \qquad \left\langle \frac{\partial L}{\partial z}, z \right\rangle = 0 \tag{7}$$

where $L$ is any loss function and $\langle \rangle$ denotes an inner product.

The proof of Equation 7 can be found in (Wang et al., 2017a). As a result, all changes to features caused by gradient adjustments to weights during the backward pass push features along their tangent line to the hypersphere. This makes features larger (the hypotenuse grows larger with side length, as per Pythagorean theorem, see Figure 3). During the subsequent forward pass features are normalized, preventing the variability in feature norms from leading to a sub-optimal loss (i.e. the hypersphere condition is still met). At the same time, any change to the direction of feature vectors is preserved.

This means that feature vectors are now "decoupled" from the usual optimization dynamics. Feature vectors (as output from the encoder, upstream from L2 normalization) are not restricted to equinormality and

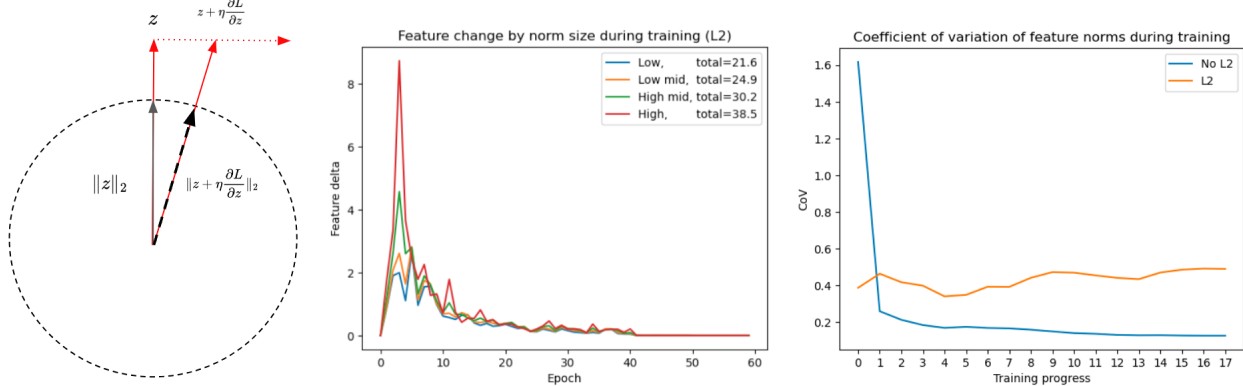

Figure 3: (Left) As a result of the orthogonality of the loss w.r.t. features (when L2 normalization is used), any weight updates that result in changes to features push features along their tangent line to the hypersphere. This means each backward pass makes features slightly larger, and this trend continues indefinitely in the absence of weight decay (Image adapted from (Wang et al., 2017a)). (Center) We separate converged features of the CIFAR10 test set into four groups, based on feature norms. As predicted, features that had the highest magnitudes at the end of training change the most during training. (Right) Variability of norms decreases during training to minimize CE loss without L2 normalization (the equinormality condition of NC), however, it increases during training with L2 normalization.

become correlated with total change due to backpropagation, as per Equation 7. Thus for our use case, contra Wang et al. (2017b), norm growth is a feature, not a bug. Intuitively, features that grow large become a measure of how much the network has had to adjust weights that affect the feature representation of an image. The theory of Coherent Gradient proposed by Chatterjee (2020) argues specifically that weight updates consistently benefit the broadest number of images: average gradient directions (from batches) are strongest where the most per-example correspondences occur, and weights are then updated proportionally to these gradient directions. This means that weight updates are biased toward handling images with the most common features (See also Chatterjee and Zielinski (2022)). This "common feature" information is then encoded in norms through L2 normalization, making norms a proxy of how familiar a given input is. Without L2 normalization, this information is largely discarded from norms due to the neural collapse that occurs during optimization under CE loss. The strength of this link between common features, weight updates and feature norm size could depend on several factors, and is not guaranteed for all models or datasets. These may include model capacity and dataset complexity, as well as anything that would affect gradient coherence, such as batch size, class imbalances or gradient accumulation in distributed training settings.

Finally, we note that $L_2$ normalization requires no hyperparameter tuning and can benefit training efficiency. We confirm that model convergence occurs faster in some architectures (We observe this for ResNet18 and ResNet50), which is in line with findings from Yaras et al. (2023) and Haas et al. (2023). Yaras et al. (2023) and E and Wojtowytsch (2020) provide theoretical rationale for why norm constraints promote convergence.

## 4 Experiments

Our results in Table 1 demonstrate that L2 normalization over feature space during training produces results that compare well with state-of-the-art methods. Notably, L2 normalization results not only in large performance gains over non-normalized baselines, but these gains happen much faster (Table 4). Training details for all experiments can be found in A.1.

### 4.1 Equinormality Under Cross Entropy Loss

The neural collapse behaviour that occurs under cross-entropy training predicts that feature norms should decrease in variability during the course of training in order to meet the equinormality conditions of optimal

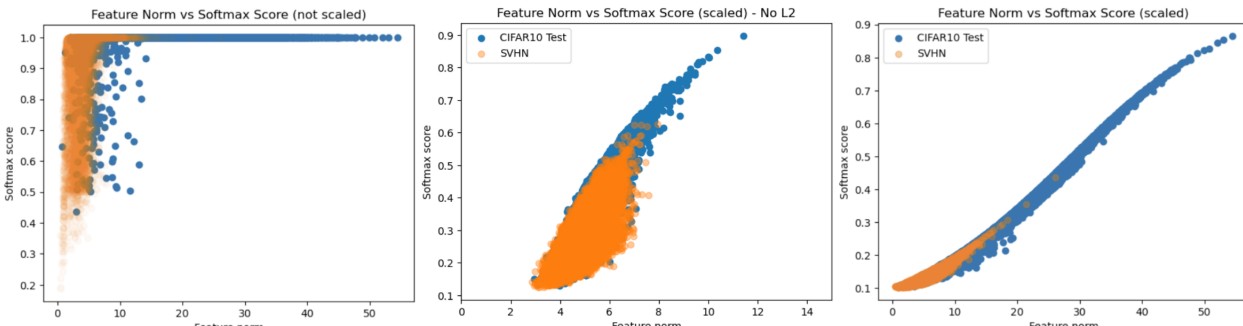

Figure 4: Plots of feature norms vs softmax scores for all CIFAR10 (blue) and SVHN (orange) test images. (Left) Allowing feature magnitudes to grow saturates the softmax function, and decreases the headroom available for separability. This occurs in both L2 and NoL2 models, although saturation is greater in L2 models. (Center) Scaled down features from the NoL2 ResNet18 350 model. (Right) Scaled down features from the L2 ResNet18 60 model. The nearly linear correlation of feature norm to softmax, in addition to the more isolated cluster of OoD images, is evidence that more image-level information is retained on a per-image basis than in NoL2 models.

loss. We observe that the variability of feature norms indeed decreases throughout training in NoL2 models (Figure 3, Right). However, in the L2 case variability increases as training progresses, since equinormality is already optimal due to normalized features and features are free to grow as described in Section 3.2.

Even though there is a clear mechanism that makes feature norms larger and more variable during training under L2 normalization, the question remains as to why OoD detection works much more poorly without it in some models and datasets. Specifically, one might speculate that even though CE loss has an equinormality condition, OoD features would still activate convolutions to a smaller degree, and thus generally result in smaller feature norms. The results in Table 4 show that this is somewhat the case: separability is above random chance still but performs poorly. We hypothesize that equinormality is achieved largely through norm invariance to inputs: the network develops convolutions in such a manner as to generate similar sized feature norms under a variety of ID and OoD inputs.

To test this, we compare feature norms from CIFAR10, SVHN, pixel-scrambled CIFAR10, and Gaussian noise. The results are shown in Figure 5. In NoL2 models, norms are almost the same average size across all datasets and variability within datasets is limited. This changes markedly in L2 models, where ID images have substantial variation amongst norms, but also much larger norms (on average) for ID data. As expected, models attempt to produce invariant norms as predicted by neural collapse. Most notably, this behaviour encourages these models to produce the same sized norms regardless of input, limiting the information encoded in features.

### 4.2 Norm Growth During Training

We first test whether Equation 7 holds empirically. In an L2 model, we expect that pre-normalized feature norms taken from a converged model would be largest when features for those images changed the most during training, due to orthogonal gradient updates.

To test this, we use a converged L2 60 model. CIFAR10 training images were separated by their converged feature norms into four categories: low in [0,10), low-mid in [10,20), high-mid in [20,30) and high in [40, ...]. These images were then run through saved model checkpoints at each epoch and the respective total changes of each group were calculated from one epoch to the next. Total change $\Delta_m$ per norm group $m$ was then calculated as the mean of the squared difference of features at consecutive epochs during training:

$$\Delta_m = \sum_i \frac{1}{N_m}(Z_{m,i} - Z_{m,i-1})^2 \tag{8}$$

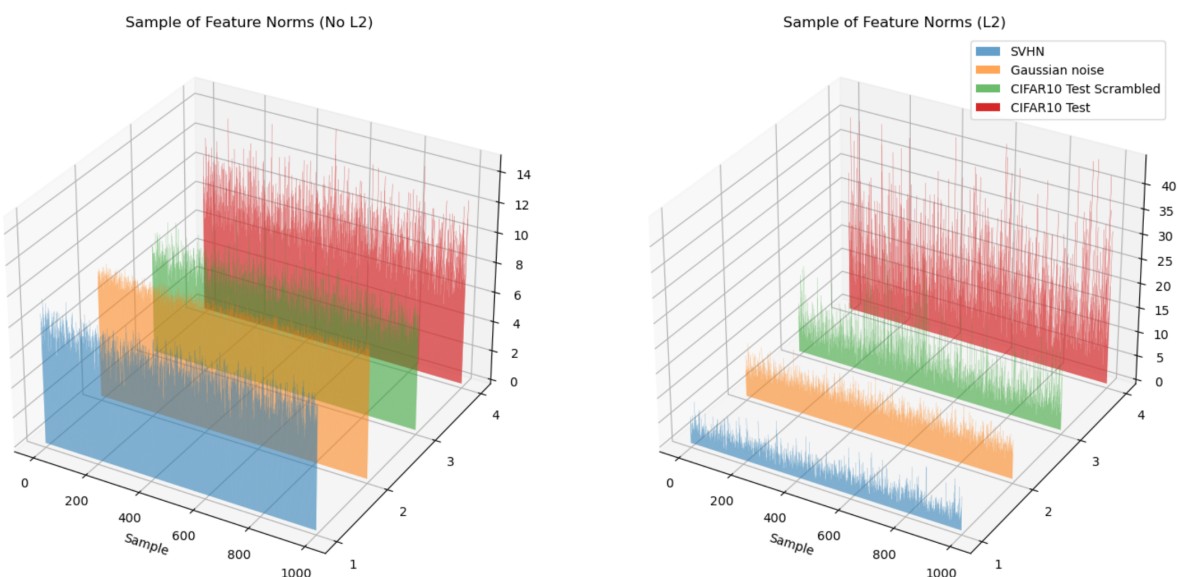

Figure 5: Comparison of norms under NoL2 ResNet18 350 (left) and L2 ResNet 60(right) models, four datasets (CIFAR10 Test, SVHN, Gaussian noise and pixel-wise scrambled CIFAR10 Test). We hypothesize that models without L2 normalization generate norms that are invariant to inputs in order to meet the equinormality condition of CE loss. Our results support this claim: all datasets have roughly equivalently sized norms on average with very little variation amongst them. When L2 normalization is used during training, ID norms (measured prior to normalization) grow large and have high variability, while OoD norms are much smaller. The sensitivity of convolutions to features is not being conditioned/suppressed to produce equinormality in the latter case, which results in richer feature-level information and better OoD detection.

where $i$ is the epoch of the model checkpoint and $Z_{m,i}$ is the matrix of $N$ feature vectors in norm group $m$ at epoch $i$.

As predicted, features with more change during training have larger magnitudes (Figure 3). We observe that this behaviour is not present when training without L2 normalization. Instead, although feature change differs slightly earlier in training, features experience roughly equal amounts of change if measured through to model convergence (Figure 6).

## 4.3 Image Information Encoded in Features

We hypothesized in Section 3 that what we call *image* information (i.e. information about the particular characteristics of an image, as opposed to information that is simply which class an image belongs to) is discarded for some architectures and datasets due to the optimization behaviour of cross-entropy loss. To assess this, we measure the relationships between feature norms and model confidence as well as feature norms and model accuracy.

To examine this in L2 models, we measure the pre-normalized features. Here, we first need to reduce the softmax saturation caused by these larger norms. This saturation arises because logits are the dot-product of features and decision layer weights – if features increase in size, logits increase correspondingly, and this results softmax scores being compressed at the high end (See (Figure 4, Left). Reducing saturation is therefore a simple, uniform downscaling of feature vectors. Note that noL2 models require no downscaling, as features were not normalized to begin with, and are thus measured in the regular state.

Plotting feature norms versus per-image maximum softmax scores, we observe that feature norms from L2 models provide a wider range of softmax scores than their counterparts without L2. Moreover, the correlation is nearly perfectly linear in the L2 case (Figure 4). This wider, more uniform softmax score coverage in the L2 case is evidence that image information is being collapsed into class information to a much smaller degree.

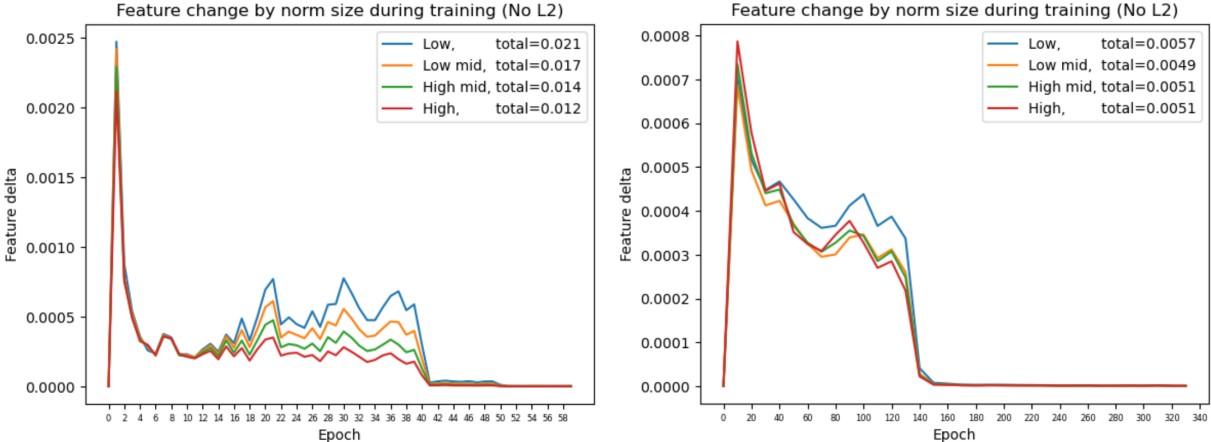

Figure 6: Feature change during training for four categories of norms in NoL2 models: low in [0,5), low-mid in [5,7), high-mid in [7,9) and high in [9, ...] (Note that these ranges differ from the L2 models as those norms are much larger). (Left) results from a NoL2 ResNet18 60 model, (Right) results from a NoL2 ResNet18 350 model. Unlike the L2 models, a larger change in features during training does not result in larger feature norms. In the first 60 epochs of training this trend is slightly reversed compared to the L2 model, but over training to full convergence (350 epochs) all norm size categories become roughly equal in the total amounts of change measured. Much more neural collapse occurs from 60 to 350 epochs, so it is likely that this equality is a because of the equinormality constraint of CE loss.

We also note improved OoD detection performance between noL2 and L2 models when using softmax scores, although these don't work quite as well as norms (Table 4, create a table with noL2 softmax baselines).

Comparisons of feature norms with prediction accuracy provide further evidence of our hypothesis. In this case, we take feature norms and group them into 125 evenly sized bins, and then calculate the average accuracy per bin (Figure 7. Accuracy tends to improve with norm size for the NoL2 350 model but the trend is very noisy, particularly with smaller norms. For the L2 case, there is a nearly monotonic relationship between norm size and accuracy. This is further evidence that feature norms contain more image information in the L2 case.

The question remains as to why exactly the amount of change to features over training correlates so strongly with both model confidence and accuracy for L2 models. We speculate some combination of 1) features that have moved around more are more likely to be closer to decision points because more effort has been invested in placing them there and 2) "common" features exist (i.e. features that provide strong class signals across many images) that have their respective weights updated more often because of their presence across so many batches. This view is supported by the Coherent Gradient theory, as noted above (Chatterjee, 2020). Intuitively, it is possible that images with low confidence also have rare and/or noisy features, and updates to the respective weights simply participate in backpropagation less often and in a less prominent manner. A causal analysis of the exact mechanism is left for future work.

## 5   Conclusion

Our results in Table 1 demonstrate that L2 normalization over feature space during training can produce OoD detection results that perform well for some architectures and datasets. We provide a theoretical rationale for this behaviour, and observe that this is due to the inclusion of more image information in feature norms than is afforded in standard CE loss training regimes. Notably, this method requires nominal additional computation, no complex architecture changes or loss functions, no extra parameter tuning and no augmentation schemes. We believe that further study into the relationships between neural collapse and normalization could lead to substantial improvements in OoD methods that employ feature vectors for OoD detection.

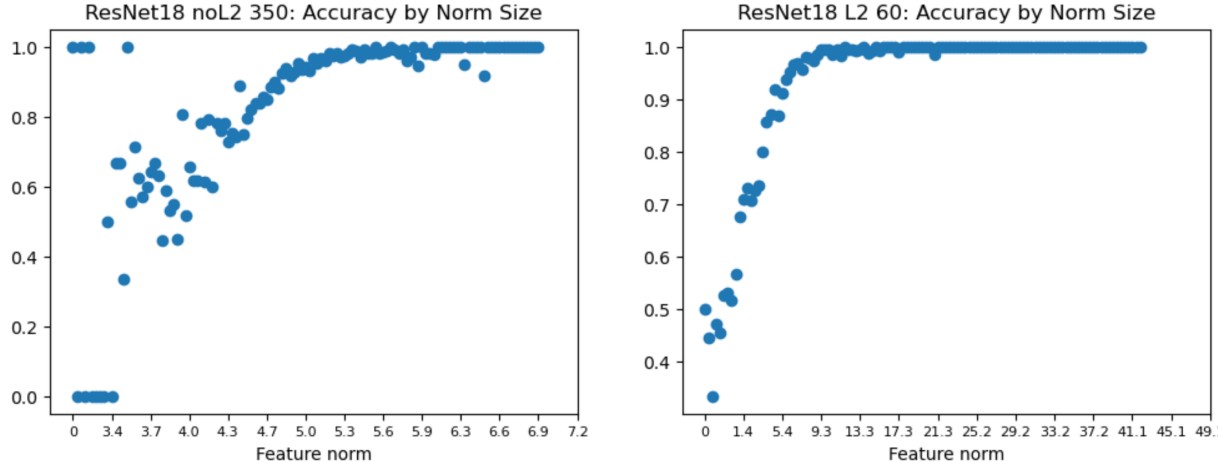

Figure 7: Feature norm vs. prediction accuracy for ResNet18 models with and without L2 normalization on CIFAR10 test data. Test images are grouped into 125 equal sized bins, and the accuracy of each bin is calculated. (Left) accuracy tends to improve with norm size for the NoL2 350 model but the trend is noisy, particularly with smaller norms. (Right) There is a nearly monotonic relationship between norm size and accuracy.

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

# A    Appendix

## A.1    Training Details

All CIFAR 10 ResNet and VGG experiments use fifteen randomly initialized models trained with Cross Entropy (CE) loss unless otherwise noted. Training employed an SGD optimizer initialized to a learning rate of 1e-1 with gamma=0.1, and with stepdowns at 40 and 50 epochs for 60 epoch models, 75 and 90 epochs for 100 epoch models, and at 150 and 250 epochs for 350 epoch models. All ResNet models use spectral normalization, global average pooling, and Leaky ReLUs, as these have shown to produce strong baseline performance (Mukhoti et al., 2021; Haas et al., 2023). We apply these strategies to LogitNorm, and demonstrate a small improvement, so we report the stronger baseline for that method which we call LogitNorm+ (Table 1). A batch size of 1024 was used wherever permitted by GPU RAM, but LogitNorm models were trained with a batch size of 128 as per the original paper's recommendations (Wei et al., 2022). ResNet50 used a batch size of 768 for CIFAR10 and 512 for TinyImageNet.

The Compact Convolutional Transformers were trained from five random initializations with cosine annealing for 300 epochs, in distributed mode parallel with batch sizes of 128. The code for these models and the training regime can be found at https://github.com/SHI-Labs/Compact-Transformers.

ConvNeXt(Tiny) was modified from its default PyTorch implementation so that it would converge better on CIFAR10. The kernel size of the first layer was lowered from 4 to 3, the stride was lowered from 4 to 2. All other kernel sizes were lowered from 7 to 3. It was trained using a single cosine annealing schedule with the AdamW optimizer.

All other baselines are reported as per their respective papers.

## A.2    Measuring Decoupled Feature Norms

| Method | SVHN | | CIFAR100 | | TinyImageNet | |
|---|---|---|---|---|---|---|
| | Norm | SM | Norm | SM | Norm | SM |
| VGG16 | 95.8 / 28.5 | 94.8 / 36.4 | 88.4 / 59.1 | 87.9 / 60.8 | 89.9 / 50.9 | 89.3 / 53.6 |
| ResNet18 | 97.4 / 14.8 | 96.8 / 18.8 | 88.7 / 55.4 | 88.7 / 55.9 | 91.1 / 44.6 | 90.7 / 47.1 |
| ResNet50 | 98.6 / 7.80 | 98.5 / 9.10 | 90.4 / 49.3 | 90.5 / 49.5 | 91.9 / 43.8 | 91.9 / 43.4 |

Table 2: Comparison of L2 models with AUROC/FPR95 scores, using feature norms and softmax as scoring rules. VGG16 and L2 ResNet18 are trained for 60 epochs, and ResNet50 is trained for 100 epochs. Scaled softmax performs comparably, but is slightly worse in general. Note that for the softmax case, norms must be scaled down to mitigate saturation, and this requires tuning the scaling parameter on test data.

| Method | SVHN | | | CIFAR100 | | |
|---|---|---|---|---|---|---|
| | Norm - Feature | Norm - Logits | MaxLogit | Norm - Feature | Norm - Logits | MaxLogit |
| Min | 96.3 / 6.93 | 95.9 / 7.51 | 95.6 / 9.12 | 88.2 / 52.1 | 88.5 / 51.0 | 88.3 / 52.6 |
| Max | 98.6 / 23.5 | 98.4 / 24.3 | 98.1 / 27.9 | 89.4 / 57.6 | 89.5 / 56.3 | 89.4 / 57.9 |
| Mean | 97.4 / 14.8 | 97.2 / 15.9 | 96.9 / 18.7 | 88.7 / 55.4 | 88.9 / 53.9 | 88.8 / 55.8 |

Table 3: Comparison of feature norm, logit norm and MaxLogit measurements over 15 ResNet18 L2 60 model seeds, for AUROC / FPR95 metrics. Using logit norms or MaxLogit provides no benefits over feature norms for near or far-OoD data.

The question of how best to measure this increased, feature-level information remains. Several papers, beginning with Hendrycks et al. (2019), report strategies based on taking the maximum logit (MaxLogit) (Vaze et al., 2022; Jung et al., 2021; Zhang and Xiang, 2023). Measuring the norm of the logits or the maximum softmax score are also options.

Softmax measurements are problematic for two reasons. First, the model is incentivized to increase weight size in the decision layer to reduce loss, as per Equation 5. This increases saturation of softmax outputs (see Figure 4), since logits are a dot product of feature vectors with the columns of decision layer weights. Secondly, in order to maximize the benefits of feature magnitudes, normalization must be turned off at test time. Substantially larger features easily saturate the softmax, removing much of the dynamic range needed to separate OoD examples. A few options exist for mitigating softmax saturation, including normalization of decision layer weights, weight decay and feature scaling.

Scaling these features down (which requires a post-hoc estimate of the optimal scaling value) produces a nearly linear correlation between softmax scores and feature magnitudes. This greatly improves OoD separation with softmax scores, but measuring magnitudes directly still gives a very small advantage (Table 2). While interesting, we don't see any practical use for this approach, and it does not improve accuracy. We also experimented with using L2 normalization over decision weights and larger weight decay values for the optimizer with the hope of mitigating saturation, but these required more training time and resulted in worse performance. In our estimation, the best approach is to measure the feature norms directly. This is the most direct source of information about the inputs. Neural collapse stipulates that the decision layer will form a dual space (up to a scalar) of the simplex ETF in feature space (Papyan et al., 2020): taking measurements from the decision layer and lower should add no substantial information, as is supported by our experimental results. Furthermore, we see no benefit to the additional complexity added by adjusting features or logits with scaling or softmax temperature terms.

### A.3 Additional Results

| Method | SVHN No L2 | SVHN L2 | CIFAR100 No L2 | CIFAR100 L2 | TinyImageNet No L2 | TinyImageNet L2 | Accuracy No L2 | Accuracy L2 |
|---|---|---|---|---|---|---|---|---|
| VGG16 | 82.4 / 72.2 | 95.8 / 28.5 | 84.4 / 63.1 | 88.4 / 59.1 | 86.4 / 57.0 | 89.9 / 50.9 | 92.4 | 92.0 |
| ResNet18 (60) | 77.8 / 76.6 | 97.4 / 14.8 | 81.0 / 66.5 | 88.7 / 55.4 | 80.5 / 67.3 | 91.1 / 44.6 | 93.7 | 92.6 |
| ResNet50 (100) | 85.8 / 51.7 | 98.6 / 7.80 | 77.7 / 66.1 | 90.4 / 49.3 | 77.8 / 64.3 | 91.9 / 43.8 | 93.8 | 94.1 |
| ResNet18 (350) | 77.8 / 76.6 | 96.0 / 25.2 | 81.0 / 66.5 | 89.7 / 53.4 | 80.5 / 67.3 | 90.4 / 46.9 | 93.7 | 93.4 |
| ResNet50 (350) | 85.8 / 51.7 | 96.1 / 24.8 | 77.7 / 66.1 | 88.9 / 54.6 | 77.8 / 64.3 | 89.3 / 50.4 | 93.8 | 94.6 |
| CCT_7_3x1 | 62.2 / 85.7 | 92.0 / 62.9 | 70.9 / 75.9 | 89.7 / 53.1 | 76.3 / 65.0 | 91.1 / 44.0 | 96.4 | 96.4 |
| ConvNeXt_Tiny* | 82.1 / 81.3 | 88.4 / 68.9 | 82.2 / 74.0 | 85.5 / 68.2 | 83.3 / 70.3 | 85.5 / 64.2 | 88.2 | 89.2 |

Table 4: Comparison of models under L2 intervention for AUROC (higher is better) / FPR95 (lower is better) scores, using feature norms as the scoring rule. L2 models produce substantially better OoD results in most cases. Bracketd numbers indicate training epochs used for the L2 case.

| Model | SVHN No L2 | SVHN L2 | CIFAR100 No L2 | CIFAR100 L2 | TinyImageNet No L2 | TinyImageNet L2 | Accuracy No L2 | Accuracy L2 |
|---|---|---|---|---|---|---|---|---|
| ResNet18 | 50.5 / 93.3 | 95.6 / 20.5 | 45.7 / 91.2 | 93.6 / 19.9 | 49.4 / 88.5 | 94.8 / 17.4 | 84.7 | 82.8 |

Table 5: Comparison of L2 vs NoL2 ResNet18 models, when both are trained on the German Traffic Sign Recognition Benchmark (GTSRB) for 60 epochs. AUROC (higher is better) / FPR95 (lower is better) scores are shown, using feature norms as the scoring rule.

| Model | SVHN No L2 | SVHN L2 | CIFAR100 No L2 | CIFAR100 L2 | Accuracy No L2 | Accuracy L2 |
|---|---|---|---|---|---|---|
| ResNet18 | 65.7 / 90.2 | 85.4 / 51.3 | 63.5 / 88.8 | 62.0 / 91.3 | 46.5 | 39.9 |
| ResNet50 | 68.5 / 79.9 | 85.9 / 45.3 | 66.2 / 86.5 | 64.8 / 91.3 | 52.5 | 49.1 |

Table 6: Comparison of L2 vs NoL2 ResNet18 models, when both are trained on TinyImageNet for 60 epochs. AUROC (higher is better) / FPR95 (lower is better) scores are shown, using feature norms as the scoring rule.

| | **SVHN** | | **CIFAR10** | | **TinyImageNet** | | **Accuracy** | |
|---|---|---|---|---|---|---|---|---|
| **Model** | No L2 | L2 | No L2 | L2 | No L2 | L2 | No L2 | L2 |
| ResNet50 | 83.0 / 63.9 | 91.1 / 52.6 | 73.2 / 84.5 | 77.1 / 80.9 | 73.4 / 82.8 | 84.2 / 66.6 | 74.2 | 74.7 |

Table 7: AUROC (higher is better) / FPR95 (lower is better) results for CIFAR100 on NoL2 ResNet50 350 and L2 ResNet50 150 models.

| | **SVHN** | | **CIFAR100** | | **TinyImageNet** | | **Accuracy** | |
|---|---|---|---|---|---|---|---|---|
| **Model** | NoL2 | L2 | NoL2 | L2 | NoL2 | L2 | NoL2 | L2 |
| ResNet18 | 65.6 / 92.9 | 97.4 / 14.8 | 77.1 / 80.6 | 88.7 / 55.4 | 80.4 / 75.4 | 91.1 / 44.6 | 89.8 | 92.6 |

Table 8: Comparison of L2 vs NoL2 ResNet18 models, when both are trained for only 60 epochs. AU-ROC/FPR95 scores are shown, using feature norms as the scoring rule. L2 normalization substantially speeds up feature learning, and also improves OoD performance. After only 60 epochs, OoD performance is better than NoL2 models trained for the full 350 epochs, and accuracy is nearly as good (see Table 4).

