# OpenReview forum: "Exploring Simple, High Quality Out-of-Distribution Detection with L2 Normalization"
_TMLR — Accepted by TMLR_

### Review · Reviewer_rPJm · 2023-11-06

**Summary Of Contributions:**

The paper proposes an out-of-distribution (OoD) detection method based on L2 feature space normalization during training. This method has the announced advantage of reduced training time overhead compared to other methods and broad applicability across architectures and tasks. The paper also provides a rationale for the connection between L2 feature normalization, neural collapse, and the usefulness of feature sizes for OoD detection.

**Audience:**

Yes

**Broader Impact Concerns:**

There are no broader impact concerns by the reviewer.

**Claims And Evidence:**

No

**Requested Changes:**

I would kindly ask the authors to address my non-minor points listed under "weaknesses" above would be. In the current form, I lack evidence for the author's main claim that their method is a "competitive, broadly applicable" OoD detection (thus falling short of the requirement that "claims made in the submission are supported by accurate, convincing and clear evidence".

**Strengths And Weaknesses:**

Strengths:
- The proposed method is simple and in principle broadly applicable (even though the later is not demonstrated in the experiments)
- The authors study the development of feature norms during training in several settings, providing insights into the relationship of feature norms, neural collapse, and OoD detection.

Weaknesses:
- According to Table 1, the proposed method is often substantial worse than baselines and not really competitive as claimed by the authors. Also, Table 1 lacks information of the accuracy of  classifiers on CIFAR10 test; L2 normalization might deteriorate the performance here, which would be undesirable. Moreover, additionally comparing to other types of regularization (e.g. weight decay, label smoothing) would be required to provide evidence that L2 normalization provides benefits beyond standard regularizers.
- OOD detection is solely evaluated on ResNet-type imaged classifiers that are trained on CIFAR10 and evaluated on SVHN and CIFAR100 outliers. This is too limited to provide evidence that L2 feature normalization is a general-purpose OoD detection method.
- "We view this tendency for norms to grow as a feature, not a bug." Growing norms have substantial negative side-effects during deployment such as preventing quantization. To the reviewer, it still is more a bug than a feature in practical settings.
- It would be helpful to further clarify differences and benefits to the concurrent work of Regmi et al. (2023), which "concurrently proposed the same method".
- Minor: A formal definition of "neural collapse" is missing
- Minor: Why is there a a bar above equation 5?
- Minor: The y-axis in Figure 3 could be changed to a logit scale since softmax score is in [0, 1]. This would help with softmax saturation. (https://matplotlib.org/stable/gallery/scales/logit_demo.html)

---

> ### Author Response · Authors · 2023-12-14
> **Thank you for your insightful feedback, our response is below**
>
> Thank you again for these thoughtful comments and for your patience in awaiting a reply during this extremely busy time of year. We have made substantial revisions to the paper in response to your requests. We are still waiting for some models to finish training and will add more results to the paper in the next couple of days, but we have just submitted a heavily revised version along with these responses that hopefully addresses your comments in the meantime.
>
> “According to Table 1, the proposed method is often substantially worse than baselines and not really competitive as claimed by the authors. Also, Table 1 lacks information of the accuracy of classifiers on CIFAR10 test; L2 normalization might deteriorate the performance here, which would be undesirable.”
>
> We’ve softened our language and claims in response to your feedback. We now use “Competitive performance” changed to “strong performance compared to…” or “compares well with”. We think being within 2% of state of the art for models with much longer training regimes or larger parameter counts is useful to the community, and we feel that our revised language throughout the paper better represents our goals and honours your feedback on this issue.
> We have tested full training runs with the method, and the numbers are comparable: 1) far OoD (i.e. svhn) drops by around 1 point, near OoD (i.e. cifar100) increases slightly and accuracy generally improves as would be expected. We will post the full results in a forthcoming revision once the other model results are finished.
>
> “Moreover, additionally comparing to other types of regularization (e.g. weight decay, label smoothing) would be required to provide evidence that L2 normalization provides benefits beyond standard regularizers.”
>
> If we understand correctly (please correct us if we don’t!), this should be resolved in the revisions we’ve made to the scope of our claims. We no longer position our method as a general purpose OoD tool or fundamental architecture consideration – we restrict our discussion to the theoretical and empirical rationale of the architectures and datasets we report on in this paper.
>
> “OOD detection is solely evaluated on ResNet-type image classifiers that are trained on CIFAR10 and evaluated on SVHN and CIFAR100 outliers. This is too limited to provide evidence that L2 feature normalization is a general-purpose OoD detection method.”
>
> We have VGG16 as well as used CIFAR100 as an in-distribution dataset in the appendix, and we’ve added TinyImageNet as an in-distribution dataset and added a Compact ViT as an architecture. We are in process of trying to train an additional architecture as well. We’re just waiting for these final model results, and then we’ll update the additional baselines in the paper in the next couple of days.
> Again, we’ve reduced our claims significantly and no longer position this method as something that would work on any dataset or architecture.
>
> “"We view this tendency for norms to grow as a feature, not a bug." Growing norms have substantial negative side-effects during deployment such as preventing quantization. To the reviewer, it still is more a bug than a feature in practical settings.”
>
> We’ve revised our language here to “for our use case, its a feature not a bug”
> We’re not completely familiar with quantization literature, but we only see this being a problem for some types of quantization? Are there some specific cases you could point out where this would be a problem?
>
> “It would be helpful to further clarify differences and benefits to the concurrent work of Regmi et al. (2023), which "concurrently proposed the same method".”
>
> We have clarified in the related work section that the primary difference is Remi et al has some additional dataset baselines, but provides no explanation/amalysis of why the method works
>
> “Minor: A formal definition of "neural collapse" is missing”
>
>  We’ve added the measurement definitions to the new methodology section
>
> “Minor: Why is there a a bar above equation 5?”
>
> This was to indicate an average of values, but we’ve fixed the notation to make it clearer
>
> “Minor: The y-axis in Figure 3 could be changed to a logit scale since softmax score is in [0, 1]. This would help with softmax saturation.”
>
> Yes we can fix this in the forthcoming revision when the additional results are finished.
>
> We hope these revisions were helpful. Please let us know if there is anything else we can address!

---

> ### Author Response · Authors · 2023-12-18
> **more revisions submitted**
>
> Thank you again for your patience, we have submitted a number of revisions, including additional models (Compact Convolutional Transformers, ConvNeXt) and datasets (TinyImageNet, German Traffic Sign Recognition Benchmark as ID) in Tables 4,5 and 6, and we look forward to hearing your feedback.

---

> ### Comment · Reviewer_rPJm · 2024-01-04
> **Thanks for your response**
>
> I would like to thank the authors for their response. However, some questions remain:
>  - "Table 1 lacks information of the accuracy of classifiers on CIFAR10 test; L2 normalization might deteriorate the performance here, which would be undesirable": Table 1 still does not contain the accuracy of classifiers on CIFAR10 test
>  - “Moreover, additionally comparing to other types of regularization (e.g. weight decay, label smoothing) would be required to provide evidence that L2 normalization provides benefits beyond standard regularizers.” It is still not clear to me which (if any) advantages  L2 normalization provides beyond standard regularizers. My understanding is that the revision is mostly toning down the claims, but for L2 normalization being relevant for TMLR, some clear understanding of differences and benefits compared to existing regularizers would be essential.
>
> Could the authors also provide a summary where in the paper new experimental results have been added? That would make it easier to review the revision.

---

> > ### Author Response · Authors · 2024-01-05
> > **Further revisions, clarification**
> >
> > "Table 1 lacks information…”
> >
> > - We apologize for the lack of clarity around Table 1, and have made substantial revisions to the table and caption. We now report CIFAR10 (ID distribution) accuracy (at least when it is reported by the original paper authors), and discuss in the caption what accuracies could be expected where this information isn’t available. We also added author names alongside method titles for clarity. We feel this table may benefit from some additional formatting, but please let us know if we’re headed in the right direction. Please note that we also include In Distribution accuracy scores in Tables 4,5,6 in the appendix.
> >
> >
> > “Moreover, additionally comparing…”
> >
> > - Although L2 normalization can be viewed as a regularizer, we view it primarily as a means to mitigate neural collapse’s equinormality constraint, which (often) leads to norms that are highly correlated with input familiarity and thus useful as an OoD scoring method. There is some evidence that it can act as a regularizer, e.g. in some models it improves accuracy slightly, but we view this as a side-benefit to the OoD detection improvements that are not found in models trained without it. However, it’s quite possible that we haven’t understood this correctly, can you clarify further?
> >
> > “Could the authors also provide a summary…”
> >
> > - Yes we are happy to provide a summary of the additional experiments added to the appendix tables immediately below. Additionally, we’ve added a complete list of revisions at the top of this openreview page for convenience.
> >   - Added compact convolutional transformer and ConvNext to results in Table 4
> >   - Added results of full training runs for ResNet18/50 for L2 models
> >   - Added results for ResNet18 with GTSRB as ID dataset for Table 5
> >   - Added results for ResNet18/50 with TinyImagenet as ID dataset for Table 6

---

> > > ### Comment · Reviewer_rPJm · 2024-01-10
> > > **Final feedback**
> > >
> > > Thanks for clarifying my questions. With the additional information, I was able to provide a recommendation for the paper.

---

> > > > ### Author Response · Authors · 2024-01-10
> > > > **Thank you**
> > > >
> > > > Thank you for your review, the paper has benefitted substantially as a result.

---

### Review · Reviewer_vCRs · 2023-11-08

**Summary Of Contributions:**

This paper discovers an interesting phenomenon that L2 normalization over feature space can produce competitive results for Out-of-Distribution (OoD) detection. The authors show both empirically and theoretically that, when Neural Collapse happens, L2 normalization preserves more feature-level information than a standard CE loss training regime, and allows greater separability between ID norms and near-OoD or far-OoD norms. Experiments with ResNets in CIFAR10 and SVHN verifies the explanation for the usefulness of L2 norm in OoD detection.

**Audience:**

Yes

**Claims And Evidence:**

No

**Requested Changes:**

1. The authors claim that they offer a theoretical rationale for the connection between L2 normalization of features, Neural Collapse and the usefulness of feature magnitudes for OoD detection. However,
- The connection relies on the Neural Collapse constraint, which may not always happen;
- The theoretical rationale has no rigorous proof;
- Why specify the training epoch to a number much smaller than the standard setting. What if one train the networks with longer epochs?

2. The experiments are limited to simple settings:
- Small datasets instead of more realistic datasets such as ImageNet;
- More popular networks such as Transformer based architectures such as ViT are not examined;
- Can the method also be compatible with self-supervised large models such as CLIP?

**Strengths And Weaknesses:**

**Strengths**

i) The phenomenon is interesting;

ii) The authors offer both theoretical and empirical discussion to explain the rationale of the phenomenon.

**Weakness**

i) The theoretical results rely on the neural collapse premise;

ii) The experiments are limited to simple settings;

---

> ### Author Response · Authors · 2023-12-14
> **Thank you for your helpful feedback, our response is below**
>
> Thank you again for these thoughtful comments and for your patience in awaiting a reply during this extremely busy time of year. We have made substantial revisions to the paper in response to your requests. We are still waiting for some models to finish training and will add more results to the paper in the next couple of days, but we have just submitted a heavily revised version along with these responses that hopefully addresses your comments in the meantime.
>
> “The connection relies on the Neural Collapse constraint, which may not always happen; The theoretical rationale has no rigorous proof;”
>
> To address these concerns, we have substantially reduced the scope of our claims. We no longer argue that L2 normalization should be considered for benefits across all architectures and/or datasets. Instead, we focus instead on its success for the architectures and datasets that we report on and relevant theoretical explanations for these cases.
> We have added multiple additional citations showing the prevalence of neural collapse under cross-entropy loss alone, as well as with regularization. Additionally, we have revised the methodology section to clarify that the equinormality constraint is not required for L2 normalization to promote norm growth in response to weight updates. Rather, L2 can create this response despite neural collapse, when it happens. According to a body of work encompassing several papers, it should happen with CE nearly all of the time to varying degrees. We agree that it is possible that, under some architectures or datasets, optimization dynamics may be such that norms contain substantial information about input familiarity without L2 normalization. We have modified our language and the claims of the paper to speak to the theoretical rationale for the specific group of architectures and datasets where this is not the case, i.e. where l2 normalization can increase the useful information present in norms, despite the equinormality constraint in those model/dataset contexts.
>
> “Why specify the training epoch to a number much smaller than the standard setting. What if one trains the networks with longer epochs?”
>
> We focus on these reduced training results to highlight the efficiency that is possible for ResNet architectures. Our lab has a limited compute budget, and models that can be efficiently trained with very capable results can extremely helpful in creating fast baselines for OoD performance. We feel this would be helpful in other settings as well. However, we can certainly add a table showing the results of full training regimes–the numbers are comparable, but 1) far OoD (i.e. svhn) drops by around 1 point, 2) near OoD (i.e. cifar100) increases slightly and 3) accuracy improves as would be expected.
>
> “Small datasets instead of more realistic datasets such as ImageNet”
>
> To address this concern, we’ve reduced our claims as mentioned above to focus on the rationale and capability of a subset of models/datasets that display these characteristics, specifically, small datasets. We are not certain to what extent this method can generalize to any model/dataset, and we have revised the paper to make this clear. The method does work on TinyImageNet, which of course is just a downsampled subset of ImageNet, and we are just waiting for more model seeds to train to retrieve the final results on this. We have had some problems replicating results on the full size imagenet, so we can’t comment on that for now.
>
> “More popular networks such as Transformer based architectures such as ViT are not examined”
>
> We were able to replicate results for compact ViTs (time and compute is limited, so we can’t comment on other varieties of transformers right now), and will be posting these results shortly. L2 normalization improves AUROC by around 18-20 points for these models on CIFAR10/SVHN, more results will be included in the final revision.
>
> “Can the method also be compatible with self-supervised large models such as CLIP?”
> We’re not sure, self-supervised models have different optimization dynamics of course, we would have to look into that case specifically.
>
> We hope these revisions were helpful. Please let us know if there is anything else we can address!

---

> ### Author Response · Authors · 2023-12-18
> **more revisions submitted**
>
> Thank you again for your patience, we have submitted a number of revisions, including additional models (Compact Convolutional Transformers, ConvNeXt) and datasets (TinyImageNet, German Traffic Sign Recognition Benchmark as ID) in Tables 4,5 and 6, and we look forward to hearing your feedback.

---

> > ### Comment · Reviewer_vCRs · 2024-01-04
> > **Thank you for the response**
> >
> > Thank you for the rebuttal (apologize for my late reply since they were not visible to me in previous weeks) and for providing lots of additional results under a limited computing budget. With the supplementary results and the pruned claims, my concerns are all addressed.

---

> > > ### Author Response · Authors · 2024-01-05
> > > **Thank you**
> > >
> > > Thank you for your insights and feedback, the paper has improved substantially.

---

### Review · Reviewer_NgcT · 2023-11-21

**Summary Of Contributions:**

They propose OOD detection by training a model with L2 normalization and using the norm of the pre-normalized features for OOD detection. They connect it to neural collapse and then give intuition why this method works. Training with L2 normalization causes norm of pre-normalized feature to get updated throughout training, so common features across more examples will get updated more and have larger magnitude.

**Audience:**

Yes

**Claims And Evidence:**

Yes

**Requested Changes:**

Major:
- I had a hard time following the key contribution of this work compared to previous works. L2 normalization during training was proposed in the previous work (if I understand correctly). Is the key difference that the previous work (https://openreview.net/pdf?id=fjkN5Ur2d6 ) only applies L2 normalization for DDU while this work applies L2 normalization for any model? Section 2 includes both the related work and the “methodology” of this work. I think the related work and “methodology” should be separate so it is more clear what is proposed in this work.
- Also, I think section 3.1 and 3.2 should be in a separate section since they give intuition of the methodology. Right now, it is presented more as an additional experiments, which I think minimizes its impact. Also, the main result of Table 1 is mentioned just in passing in Section 3. I think they should be emphasized more as evidence of why this method is good - works well and is simple.
- Overall, I thought the contributions listed were scattered with lots of additional points and it's not always clear how these additional points related to the contributions, thus underemphasizing the contributions.

Minor:
-  "Given that the most confident predictions have the largest feature norms after training"
Is this because the dot product of the feature and the class representation is high for confident predictions?
I saw the reasoning later in the experimental section “ In the absence of L2 normalization during training, larger features mean a larger dot product with the decision layer, and thus a larger softmax score” - I think it should be moved earlier
-  "unclear why the amount of change to features over training correlates so strongly with model confidence and accuracy"
Is the amount of change to features correlated with the magnitude of the feature which is then correlated with model confidence?
- “We speculate some combination … “
Is this speculation to explain why change in features correlates with model confidence?
Also, this paper (https://openreview.net/pdf?id=ryeFY0EFwS) seems to get at a similar explanation with some justification.
- In figure 3 for the center and right plots, where are the isolated clusters of OOD images?

**Strengths And Weaknesses:**

Strength:
Overall, I liked the method being proposed and the intuition for it. The method is simple, which makes it easier to use and thus more impactful. Also, the intuition for the method was nice, connecting it to neural collapse and then motivating why L2 normalization is beneficial. I am not that familiar with this area thought, so I am not sure of the novelty of this.

Weakness:
The presentation of this paper makes it hard to follow what I thought were the strengths of this paper. My impression of this paper got better as I understood is more, whereas normally when I read a paper, it is the opposite. Thus, I think this paper could be greatly improved with better presentation so readers could more easily understand its strengths. See changes below.

---

### Decision · Action_Editor_8txs · 2024-01-25

**Recommendation:** Accept as is

**Comment:**

The authors propose an OOD detection method based on L2 normalization over feature space.

All three reviews after the rebuttal, recommending acceptance advocating for merit of simplicity (yet effectiveness) of the method. While the method does not reach SoTA performance on the benchmarks, the suggested simple L2 normalization method has potential to enlighten what representation is learned during deep learning training on the dataset.

"The additional experiments and the revisions to the claims addressed my concerns" reviewer `vCRs`

" main contribution of this paper is the motivation of using L2 normalization (connecting it to neural collapse), its simplicity, …" reviewer `NgcT`

**Audience:**

The paper proposes a simple method for Out-of-Distribution detection using L2 normalization over feature space. The proposed method is simple and broadly applicable. The authors also provide insights into relationships among feature norms, neural collapse and OoD detection which may be of interest to a wider TMLR audience .

"...it still could be interesting for a part of TMLR's audience." reviewer `rPJm`

**Claims And Evidence:**

There are some concerns whether the claims are fully supported by experiments of the paper.
Mainly the proposed method does not produce outright SoTA performance on OoD detection, however reviewers still appreciated the simplicity of the method.

Reviewers pointed out that some of the strong initial claims were not fully supported.  This has been resolved during the rebuttal phase to address reviewer concerns. Revised submission focuses on simplicity of the method.

"Where no sufficient evidences were provided, the claims have been toned down." - reviewer `rPJm`

"... the main benefit of this method is its simplicity"  - reviewer `NgcT`